# Pattern and perceived changes in quality of life of Vietnamese medical and nursing students during the COVID-19 pandemic

**Le Dai Minh**[1], **Hoang Huy Phan**[1], **Duong Ngoc Le Mai**[1], **Nguyen Tien Dat**[1], **Ngo Minh Tri**[1], **Nguyen Viet Ha**[1], **Nguyen Huu Tu**[2], **Kirsty Foster**[3], **Kim Bao Giang**[4], **Tung Thanh Pham**[5]*

**1** Doctor of Medicine Program, Hanoi Medical University, Hanoi, Vietnam, **2** Department of Anesthesia, Hanoi Medical University, Hanoi, Vietnam, **3** Academy for Medical Education, Faculty of Medicine, University of Queensland, Brisbane, Australia, **4** School of Preventive Medicine and Public Health, Hanoi Medical University, Hanoi, Vietnam, **5** Department of Physiology, Hanoi Medical University, Hanoi, Vietnam

* phamthanhtung@hmu.edu.vn

## Abstract

### Background

The COVID-19 pandemic and governments' response lead to dramatical change in quality of life worldwide. However, the extent of this change in Vietnamese medical and nursing students has not been documented.

### Objectives

The study aims to describe the quality of life and changes in quality of life of medical and nursing students during the COVID-19 pandemic and examine the association of quality of life and changes in quality of life with fear of COVID-19 and other socio-economic and demographic factors.

### Methods

The study was a cross-sectional study on all students of Hanoi Medical University from 3 majors: General Medicine, Preventive Medicine, Nursing (3672 invited students); from 7th to 29th of April 2020; using an online questionnaire that included demographic and academic information, the Vietnamese version of the SF-36 Quality of Life questionnaire and the Fear of COVID-19 Scale (FCV-19S). Linear and modified Poisson regression was used to examine the association between quality of life, changes in quality of life and other factors.

### Results

The number of participants was 1583 (response rate 43%). Among 8 dimensions of the SF-36 (ranged 0–100), Vitality had the lowest score with a median score of 46. The median physical composite score (PCS) of the sample was 40.6 (IQR:20.8–53.2), 33.5% of the sample had an above-population average PCS score. The median mental composite score

**Data Availability Statement:** According to our application to the Institutional Review Board (IRB) of Hanoi University of Public Health, the data cannot be shared publicly because of ethical

restrictions to protect the confidentiality of the participants imposed by the IRB. A de-identified dataset is available for researchers who meet the criteria for access to confidential data. Requests for data should be submitted to Institutional Review Board of Hanoi University of Public Health (irb@huph.edu.vn) and the corresponding author, Dr. Pham Thanh Tung (phamthanhtung@hmu.edu. vn).

**Funding:** The author(s) received no specific funding for this work.

**Competing interests:** The authors have declared that no competing interests exist.

(MCS) of the sample was 20.3 (IQR:3.8–31.7), and 98.2% had an MCS score below average. 9.9% (95%CI:8.5%–11.4%) of the population reported a significant negative change in the quality of life. Fear of COVID-19 was not associated with significant changes in quality of life, nor MCS while having some association with PCS (Coef:-5.39;95%CI:-3to-7.8). Perceived reduction in quality of life was also associated with: being on clinical rotation COVID-19 (PR:1.5;95%CI:1.05–2.2), difficulties affording health services (PR:1.4;95%CI:1.02–1.95), obesity (PR:2.38;95%CI:1.08–5.25) and chronic disease (PR:1.92;95%CI:1.23–3), typical symptoms (PR:1.85; 95%CI:1.23–2.78) and atypical symptoms of COVID-19 (PR:2.32;95%CI:1.41–3.81).

## Conclusion

The majority of medical and nursing students had below average quality of life, with lower vitality and mental composite health score in the settings of COVID-19. Perceived decrease in quality of life was associated with clinical rotation, difficulties affording healthcare services and was not associated with Fear of COVID-19.

## Introduction

The COVID-19 pandemic caused by the novel coronavirus SARS-CoV-2, emerged in Wuhan, China in December 2019 [1]. Since then, the outbreak had progressed rapidly across the globe with a high death count and devastating impact on multiple aspects of life in many countries, including physical and mental health, politics, economy, and many other social issues [2–8]. Healthcare workers were shown to have a significantly higher risk of COVID-19 infection, compared to the general public, and faced with extraordinary amounts of pressures leading to physical and mental exhaustion [9–14]. Medical and nursing students, especially those that are on clinical rotation, are being exposed to a similar environment and risk for COVID-19 infection.

Medical and nursing students, before the pandemic, were already known to face multiple physical and mental problems including burnout, anxiety, depression, and other mental health issues, with stress, lack of academic motivation, and financial hardship being important risk factors [15–21]. During the COVID-19 pandemic, students at higher education institutions, including medical students, greatly suffer from increased depressive symptoms, anxiety and sleep disturbance [22, 23]. Medical education also changed rapidly in response to the situation, with the replacement of in-person classes with online equivalents and disruption of clinical rotations [24–29]. In Vietnam and many other countries, medical and nursing students also serve in the front line, supporting attempts to control the disease including: contact tracing, providing counseling via telephone hotline, data management, cleaning/disinfecting, and taking test samples from suspected cases [30]. Together with other socio-economic aspects related to the pandemic like lockdown, social distancing, limitation in physical activities, the authors hypothesized that the physical and mental quality of life of medical students could be at high risk and can potentially lead to poorer health outcomes.

Understanding changes in medical students' quality of life is necessary to implement coping strategies and policies to reduce the burden that medical students are facing in the context of COVID-19. Therefore, the authors conducted this study to (1) describe the quality of life of medical and nursing students during the social distancing period and (2) examine the

association between quality of life, fear of COVID-19, and other socio-economics and demographics characteristics.

# Methods

## Study design and setting

A cross-sectional study is conducted on 1583 students who major in medicine, preventive medicine, and nursing, at Hanoi Medical University (out of total 3672 students), from 7th to 29th of April 2020. This period is within the first 6 months of the COVID-19 pandemic in Vietnam.

Hanoi Medical University is one of the largest and most prestigious public medical schools in Vietnam. Every year, the enrollments for each major are about 400–500 for the general medicine program, 80 for the preventive medicine program and 140 for the nursing program. Students studying medicine and preventive medicine have a 6-year curriculum and are trained in general medicine (for general medicine), in preventive medicine (for preventive medicine) after they graduated, while nursing programs have a 4-year curriculum. Students studying general medicine and preventive medicine start their clinical rotation in the second semester of their third year, whereas nursing student start their rotation in the second semester of their second year. The detailed information on Vietnamese medical and nursing education can be found in other articles [31, 32].

During first 6 months of the COVID-19 pandemic in 2020, Vietnam underwent a temporary shutdown and implemented a social distancing policy [33–37]. The disease was relatively controlled as the number of cases and mortality rate were rather small [33–36, 38]. However, due to the policies, most conventional medical education activities, including clinical rotation were stopped and replaced with online education.

## Survey instruments

The questionnaire included questions regarding *demographic information*, *academic information* and *health information*, *the Fear of COVID-19 Scale (FCV-19S)*, and the *Quality of life SF-36 version 2.0 (SF-36)*, and *a question asking about the change in the general quality of life before and during the pandemic*.

The demographic information included: age, gender (Male/Female), marital status (Married/Single), perceived affordability of healthcare services (No difficulties/Difficulties). The academic information included: major (Doctor of Medicine/Doctor of Preventive Medicine/Nursing), academic year (First year/Second year/Third year/Fourth year/Fifth year/Sixth year), currently being on clinical rotation (Yes/No). Finally, the health information included: BMI (kg/m$^2$), history of chronic disease (Yes/No) and current symptoms of COVID-19 (no symptoms, typical symptoms, atypical symptoms).

In our study, *the symptoms of COVID-19* were divided into 3 categories: no symptoms, typical symptoms, atypical symptoms; with typical symptoms being: fever, cough and dyspnea and atypical symptoms being fatigue, muscle aches, sputum, anxiety, headache, sore throat, congestion, chest pain, hemorrhage, nausea and vomiting. At the time of data collection, this categorization was used in Vietnam by the Ministry of Health to screen for COVID-19 [39]. BMI was classified into 4 categories according to the WHO Asian–Pacific cutoff point: underweight (<18.5 kg/m2), normal weight (18.5–22.9 kg/m2), overweight (23–24.9 kg/m2), and obese (≥25 kg/m2) [40].

*Fear of COVID-19* was assessed using the FCV-19S questionnaire. The FCV-19S questionnaire was translated into Vietnamese in a previous study by Nguyen and colleague and showed good item-scale convergent validity (mean of Rho = 0.77), discriminant validity and construct

validity and high internal consistency (Cronbach's alpha = 0.90) with the Vietnamese translation [41, 42]. It consisted of 7 items and utilized a 5-point Likert scale with 1 = "strongly disagree", 2 = "disagree", 3 = "neutral", 4 = "agree", 5 = "strongly agree". The total score ranged from 7 to 35, with a higher score indicating greater fear of COVID-19. We used a cut-off point of 21 with scores ranging from 7–20 categorized "Low FCV-19S score", implying lower fear of COVID-19 and scores ranging from 21–35 categorized "High FCV-19S score", implying higher fear of COVID-19 as we assumed that those who answered neutrally at every question would be having a score of 21.

The *quality of life of students* during the period was determined using the Vietnamese version of the Quality-of-life SF-36 version 2.0 [43]. The SF-36 determine the quality of life using 8 dimensions: Physical functioning (PF), Role limitations due to physical health (RP), Role limitations due to emotional problems (RE), Vitality (VT), Emotional well-being (MH), Social functioning (SF), Pain (BP), General health (GH). The score for the 8 dimensions of the SF-36 ranged from 0–100 and positively correlate with the state of quality of life. A physical composite score (PCS) and mental composite score (MCS) were calculated from the 8 dimensions using a population-based scoring method [44, 45]. The two summary scores, PCS and MCS, use the sum of the eight dimension z-scores derived from a reference population of Vietnamese people in the study by Watkins in 2000 [43], weighted by factor score coefficients derived from US 1990 general population estimates [44]. A PCS/MCS score of 50 represents the reference population average, and any score below 50 would be considered "Below the population average". The PCS and MCS scores with z-score derived from USA population averages is also included in the supplemental documents (**S8 Table in S1 File**). The Vietnamese version of the SF-36 was validated and was shown to provide a valid assessment of self-reported health status among the Vietnamese population, with all but two of the 36 items displayed good discriminant validity and all eight scale had good discriminant validity displayed the internal consistency of the 8 scales: PF (0.82), RP (0.86), BP (0.58), GH (0.66), VT(0.56), SF (0.67), RE (0.70), MH (0.55) [43].

*Perceived reduction in quality of life* was evaluated with a question asking the students "How did your quality of life change compared to before the pandemic?" with the answer either "Worse" "Better" or "No difference". Students that answered "Worse" were categorized as having their quality of life worsen while other students were categorized as having no change or improvement in their quality of life.

## Sample size and data collection

The required sample size was estimated by using the formula for estimating sample mean [46], as follows:

$$n = \frac{Z_{1-\frac{\alpha}{2}}^2 * \sigma^2}{d^2} = \frac{1.96^2 * 12.5^2}{1^2} = 600.25$$

With, n being the required sample size, Z being the standard error associated with the chosen level of confidence 5%, σ being the standard deviation of the general quality of life score taken from the population used to validate the Vietnamese version of the SF-36 [43].

From April 7th to 29th, 2020, all student (3672 students) from the Doctor of General Medicine, Doctor of Preventive Medicine, and Bachelor of Nursing at Hanoi Medical University were invited to participate in this survey. 1032 students from the Doctor of General Medicine program (2921 invited, response rate 35.3%), 308 from the Doctor of Preventive Medicine program (466 invited, response rate 66.1%), and 243 from the Bachelor of Nursing program (285

invited, response rate 85.3%) agreed to participate in the study and completed the research questionnaire (1583/3672, resulting in a response rate of 43.1%.)

We collected data using a self-reported, online Google Form questionnaire, that included the study instruments. The research staff asked the Office for Student Services to send an official university notice with the questionnaire link to representatives of all invited classes. The class representatives are students that are responsible for major communication between the school administration and the student of the class that they're in. There is one class representative for each class, and this practice is common for most universities in Vietnam. Each representative then delivered the documents to their class's social media groups. Through these channels, all students in invited classes could read the information regarding the study and decide whether to participate and complete the survey. The research staff also checked the responses and sent a reminder to classes with low response rate every week. At the end of April 2020, we closed the questionnaire link and extracted the data to an Excel file, which was then converted into a Stata data file.

## Data analysis

Stata 15.1 was used to analyze data. The Chi-squared, Fisher exact, t-test, and Kruskal-Wallis test were used to compare the difference between groups of students regarding demographic characteristics, quality of life and changes in quality of life of participants. We consider a p -value $< 0.05$ as statistically significant for all statistical tests [47, 48].

Linear regression was used to determine the association between PCS and MCS score with demographic characteristics and fear of COVID-19 in students. Residual plots were used to check for linear model assumptions. The prevalence of people with changes in quality of life was high, thus, using logistic regression analysis to determine the association between the decrease in quality of life and independent variables may result in an overestimation of the relationship [49, 50]. Therefore, we used a modified Poisson regression model with a robust error variance with binary data to directly calculate the Prevalence ratio (PR) as an appropriate alternative [49, 50].

## Ethical issue

Our survey was approved by the Institutional Review Board of Hanoi Universities of Public Health and the administrative board of Hanoi Medical University (IRB No. 133/2020/ YTCC-HD3). The online questionnaire provided participants with comprehensive information about the study. All participants gave informed consent by clicking "I agree to participate" box on the informed consent page. This consent procedure was approved by the IRB. Moreover, the participants could stop participating and close the Google Form at any time during the survey.

## Results

### Sample characteristics

The response rate for the study was 43% (1583/3672). The cohort characteristics are presented in **Table 1.** Of the 1583 participants, 65% were studying General Medicine, 19.5% were studying Preventive Medicine and 15.4% majored in Nursing. The proportion of participants currently on clinical rotation was 65.4%. There are several differences among the three cohorts regarding gender, age, affordability of healthcare services, BMI, as well as the prevalence of symptoms of COVID-19 and FCV-19S scores. The proportion of females was greater among Nursing (96.7%) and Preventive Medicine (75%) students than General Medicine (50.9%)

**Table 1. Sample characteristics.**

| Columns by: Academic majors | General Medicine | Preventive Medicine | Nursing | Total | P-value |
|---|---|---|---|---|---|
| n (%) | 1032 (65.2) | 308 (19.5) | 243 (15.4) | 1583 (100.0) | |
| **Age, mean (sd)** | 21.74 (1.95) | 22.53 (1.83) | 20.10 (1.47) | 21.64 (2.00) | **<0.01** |
| **Gender, n (%)** | | | | | |
| Female, n (%) | 525 (50.9) | 231 (75.0) | 235 (96.7) | 991 (62.6) | |
| Male, n (%) | 507 (49.1) | 77 (25.0) | 8 (3.3) | 592 (37.4) | **<0.01** |
| **Academic years, n (%)** | | | | | |
| First year, n (%) | 147 (14.2) | 38 (12.3) | 97 (39.9) | 282 (17.8) | |
| Second year, n (%) | 247 (23.9) | 19 (6.2) | 79 (32.5) | 345 (21.8) | |
| Third year, n (%) | 153 (14.8) | 36 (11.7) | 37 (15.2) | 226 (14.3) | |
| Fourth year, n (%) | 103 (10.0) | 54 (17.5) | 30 (12.3) | 187 (11.8) | |
| Fifth year, n (%) | 122 (11.8) | 74 (24.0) | 0 (0.0) | 196 (12.4) | |
| Sixth year, n (%) | 260 (25.2) | 87 (28.2) | 0 (0.0) | 347 (21.9) | **<0.01** |
| **Currently on clinical rotation, n (%)** | | | | | |
| No, n (%) | 394 (38.2) | 57 (18.5) | 97 (39.9) | 548 (34.6) | |
| Yes, n (%) | 638 (61.8) | 251 (81.5) | 146 (60.1) | 1035 (65.4) | **<0.01** |
| **Marital status, n (%)** | | | | | |
| Single, n (%) | 1026 (99.4) | 304 (98.7) | 242 (99.6) | 1572 (99.3) | |
| Married, n (%) | 6 (0.6) | 4 (1.3) | 1 (0.4) | 11 (0.7) | 0.35 |
| **Affordability of healthcare services, n (%)** | | | | | |
| No Difficulties, n (%) | 543 (52.6) | 179 (58.1) | 101 (41.6) | 823 (52.0) | |
| Difficulties, n (%) | 489 (47.4) | 129 (41.9) | 142 (58.4) | 760 (48.0) | **<0.01** |
| **BMI categories, n (%)** | | | | | |
| Underweight, n (%) | 168 (16.3) | 75 (24.4) | 79 (32.5) | 322 (20.3) | |
| Normal, n (%) | 682 (66.1) | 191 (62.0) | 154 (63.4) | 1027 (64.9) | |
| Overweight, n (%) | 163 (15.8) | 38 (12.3) | 8 (3.3) | 209 (13.2) | |
| Obese, n (%) | 19 (1.8) | 4 (1.3) | 2 (0.8) | 25 (1.6) | **<0.01** |
| **Having chronic disease, n (%)** | | | | | |
| No, n (%) | 961 (93.1) | 281 (91.2) | 226 (93.0) | 1468 (92.7) | |
| Yes, n (%) | 71 (6.9) | 27 (8.8) | 17 (7.0) | 115 (7.3) | 0.53 |
| **Symptoms of COVID-19, n (%)** | | | | | |
| No symptoms, n (%) | 866 (83.9) | 244 (79.2) | 172 (70.8) | 1282 (81.0) | |
| Typical Symptoms, n (%) | 110 (10.7) | 49 (15.9) | 43 (17.7) | 202 (12.8) | |
| Atypical Symptoms, n (%) | 56 (5.4) | 15 (4.9) | 28 (11.5) | 99 (6.3) | **<0.01** |
| **FCV-19S score categories, n (%)** | | | | | |
| Low, n (%) | 756 (73.3) | 221 (71.8) | 135 (55.6) | 1112 (70.2) | |
| High, n (%) | 276 (26.7) | 87 (28.2) | 108 (44.4) | 471 (29.8) | **<0.01** |
| **FCV-19S score, median (iqi)** | 16.00 (13.00; 21.00) | 16.00 (12.25; 21.00) | 19.00 (15.00; 21.00) | 16.00 (13.00; 21.00) | **<0.01** |

Statistical comparison Statistical comparison using Chi-square test for categorical variable—display as n(%);T test for continuous-normally distributed variable—display as mean(sd); Wilcoxon rank-sum test for continuous-skewed variable—display as median(iqi); The bold p-value indicated statistical significance (p<0.05).

students (p<0.001). Among nursing students, typical symptoms of COVID-19 were more prevalent at around 17.7% (p<0.001), compared to 10.7% in general medicine and 15.9% in preventive medicine. Additionally, nursing major students also had higher scores in the FCV-19S (Median: 19.00; IQR: 15.00–21.00; p<0.01), and had more students scored "High" on FCV-19S categories (44.4%, p<0.01), compared to students from other majors.

### Description of the SF-36 scores

Table 2 describe the median (IQR) the scoring on PCS, MCS and 8 dimensions of the SF-36 in different groups.

**Eight dimensions of the SF-36.** The median score for the dimension of the SF-36 were PF = 90, RP = 100, RE = 100, VT = 46, MH = 64, SF = 75, BP = 100, and GH = 71.3 with Table 2 described the 8 dimensions of SF-36 score by major. There was no difference in VT, GH between majors and gender. However, nursing students have lower PF, RP, RE, MH, SF, BP scores compared to Preventive Medicine and General Medicine students (p<0.001). Females have lower physical functioning (PF), more role limitation due to physical functioning (RP) and bodily pain (BP) compared to males.

**Physical composite score.** The median PCS score of the sample was 40.6 (IQR: 20.8–53.2). There was a significant difference in the PCS score between the 3 majors (p<0.001). The PCS score of general medicine, preventive medicine and nursing students were 43.2 (IQR: 22.7–53.8), 39.9 (IQR: 21.3–52.5) and 29.3 (IQR: 12.5–48.6) respectively. PCS score was also higher in males than females (p = 0.005), in students in later years of the program (from 4th year) than those in earlier years (p<0.001), in students on clinical rotation than those not on rotation (p<0.001), and in students who had no difficulties affording healthcare services than those with difficulties (p<0.001). The PCS score was significantly lower in those who had symptoms of COVID-19 (p <0.001) and those who have significant fear of COVID-19 (p <0.001), compared to those who didn't. There was no difference in PCS scores regarding chronic diseases status. Overall, 66.5% (95% CI: 64.1%– 68.8%) of the sample had PCS score below the reference population average. Nursing major has the highest number of students who had a below average score (77.4%; p <0.01) (**S1 and S2 Tables in S1 File).**

**Mental composite score.** The median MCS score of the sample was 20.3 (IQR: 3.8–31.7). Similar to the PCS score, there was a difference between 3 majors (p<0.001) with the MCS score of 20.1 (IQR: 3.3–31.3) for general medicine, 22.5 (IQR: 8–34.4) for preventive medicine and 17.3 (IQR: 2.8–29.9) for nursing. The MCS score was also higher in students who had no difficulties affording healthcare services than those with such difficulties (p<0.001), in those who did not have symptoms of COVID-19 than those with symptoms (p<0.001), and in those who had insignificant fear of COVID-19 than those with significant fear (p<0.001). Students having chronic disease also had a lower MCS score, compared to those who did not (p = 0.007). On the other hand, there was no difference in the MCS score regarding gender, academic year, clinical rotation. Compared to the reference population average, 98.2% (97.5–98.8) of the sample have a lower MCS score (**S1 and S2 Tables in S1 File).**

### Regression analysis of PCS and MCS scores

We noticed no specific pattern in the residual analysis of the two linear regression models of PCS and MCS score (**S3, S4 Figs in S1 File).**

In Table 3, there was no difference in PCS scores between gender after accounting for other socio-demographic and academic characteristics, the FCV-19S scale, and symptoms of COVID-19. Chronic disease was also not associated with PCS score. Lower PCS score was associated with: majoring in Preventive Medicine (Coef: -3.3; 95% CI: -6.2 to -0.43) or Nursing (Coef: -5.67; 95% CI: -9 to -2.4), significant fear of COVID-19 (Coef: -5.39; 95% CI: -3 to -7.8) and typical (Coef: -4.96; 95% CI: -8.2 to -1.7) and atypical (Coef: -11.52; 95% CI: -16 to -7) symptoms of COVID-19, difficulties affording healthcare services were associated with PCS (Coef: -6.32; 95% C: -8.5 to -4.1). Higher PCS score was associated with being on clinical rotation (Coef: 8.89; 95% CI: 6.6 to 11). The factors with the highest influence on PCS score was being on clinical rotation (Coef: 8.89; 95% CI: 6.6 to 11) and having atypical symptoms (Coef:

**Table 2. PCS, MCS, and scoring of the 8 dimensions of the SF-36.**

| | Physical Composite Score (PCS) | Mental Composite Score (MCS) | Physical functioning (PF) | Role limitations due to physical health (RP) | Role limitations due to emotional problems (RE) | Vitality (VT) | Emotional well-being (MH) | Social functioning (SF) | Pain (BP) | General health (GH) |
|---|---|---|---|---|---|---|---|---|---|---|
| Total | 40.6 (20.8–53.2) | 20.3 (3.8–31.7) | 90 (70–100) | 100 (50–100) | 100 (33.3–100) | 46 (36–56) | 64 (52–76) | 75 (62.5–87.5) | 100 (80–100) | 71.3 (58.8–77.5) |
| **Academic majors** | | | | | | | | | | |
| General Medicine | 43.2 (22.7–53.8) | 20.1 (3.3–31.3) | 90 (70–100) | 100 (50–100) | 100 (33.3–100) | 46 (36–56) | 64 (52–76) | 75 (62.5–87.5) | 100 (80–100) | 71.3 (58.8–77.5) |
| Preventive Medicine | 39.9 (21.3–52.5) | 22.5 (8–34.4) | 90 (70–100) | 100 (50–100) | 100 (33.3–100) | 46 (36.3–56.3) | 64 (52–76) | 75 (62.5–100) | 100 (80–100) | 71.3 (58.8–81.3) |
| Nursing | 29.3 (12.5–48.6) | 17.3 (2.8–29.9) | 85 (60–95) | 75 (25–100) | 100 (33.3–100) | 46 (36–56) | 60 (52–72) | 62.5 (50–87.5) | 100 (80–100) | 71.3 (58.8–77.5) |
| P value | **<0.001** | **0.009** | **0.001** | **<0.001** | **0.026** | 0.474 | **0.004** | **<0.001** | **0.007** | 0.417 |
| **Gender** | | | | | | | | | | |
| Female | 38.8 (20.8–51.9) | 19.8 (3.7–31.4) | 90 (70–100) | 100 (50–100) | 100 (33.3–100) | 46 (36–56) | 64 (52–72) | 75 (62.5–87.5) | 100 (80–100) | 71.3 (58.8–77.5) |
| Male | 45.1 (21.2–54.2) | 21.1 (4.3–32.2) | 95 (65–100) | 100 (50–100) | 100 (66.7–100) | 46 (38.5–56.3) | 64 (52–76) | 75 (62.5–87.5) | 100 (80–100) | 71.3 (58.8–81.3) |
| P value | **0.005** | 0.288 | **0.004** | **0.031** | 0.077 | 0.105 | 0.250 | 0.375 | **0.004** | 0.417 |
| **Academic years*** | | | | | | | | | | |
| First year | 27.8 (11.1–44.1) | 16.6 (0.7–32.5) | 80 (55–95) | 75 (25–100) | 66.7 (33.3–100) | 46.3 (36–56) | 64 (52–72) | 62.5 (50–87.5) | 100 (80–100) | 71.3 (58.8–77.5) |
| Second year | 39.3 (18.1–51.5) | 20.3 (5.5–31.8) | 90 (65–100) | 100 (50–100) | 100 (66.7–100) | 46 (36.3–51.5) | 60 (52–72) | 75 (62.5–87.5) | 100 (80–100) | 71.3 (58.8–77.5) |
| Third year | 39.5 (20.7–53.2) | 19.9 (3.2–29) | 90 (70–100) | 100 (50–100) | 100 (66.7–100) | 45.8 (36–51) | 60 (52–68) | 75 (62.5–87.5) | 100 (80–100) | 65 (52.5–77.5) |
| Fourth year | 47.6 (30.2–54.3) | 21.5 (6.8–32.4) | 95 (85–100) | 100 (75–100) | 100 (66.7–100) | 46 (36–56.3) | 60 (52–76) | 75 (62.5–87.5) | 100 (90–100) | 65 (58.8–77.5) |
| Fifth year | 46.7 (31.3–55.7) | 23 (8.7–32.3) | 95 (80–100) | 100 (100–100) | 100 (66.7–100) | 46 (40.8–56.3) | 68 (52–76) | 75 (62.5–100) | 100 (80–100) | 71.3 (58.8–83.8) |
| Sixth year | 44.8 (25–54.2) | 20.6 (0.3–33.2) | 95 (75–100) | 100 (50–100) | 100 (33.3–100) | 50.3 (36–56.3) | 68 (56–80) | 75 (62.5–87.5) | 100 (90–100) | 71.3 (58.8–83.8) |
| P value | **<0.001** | 0.247 | **<0.001** | **<0.001** | **<0.001** | **<0.001** | **<0.001** | **<0.001** | **<0.001** | **<0.001** |
| **Currently on clinical rotation** | | | | | | | | | | |
| No | 32.4 (13.6–49.4) | 18.4 (2.8–31.8) | 85 (55–95) | 87.5 (25–100) | 100 (33.3–100) | 46 (36–56) | 60 (52–72) | 75 (62.5–87.5) | 100 (80–100) | 71.3 (58.8–77.5) |
| Yes | 44.5 (24.1–54.1) | 21 (5.2–31.6) | 95 (75–100) | 100 (50–100) | 100 (33.3–100) | 46 (36–56) | 64 (52–76) | 75 (62.5–87.5) | 100 (80–100) | 71.3 (58.8–81.3) |
| P value | **<0.001** | 0.316 | **<0.001** | **<0.001** | **<0.001** | 0.295 | **0.029** | **<0.001** | **0.001** | 0.692 |
| **Marital status** | | | | | | | | | | |

*(Continued)*

**Table 2.** (Continued)

| | Physical Composite Score (PCS) | Mental Composite Score (MCS) | Physical functioning (PF) | Role limitations due to physical health (RP) | Role limitations due to emotional problems (RE) | Vitality (VT) | Emotional well-being (MH) | Social functioning (SF) | Pain (BP) | General health (GH) |
|---|---|---|---|---|---|---|---|---|---|---|
| Single | 40.6 (20.8–53.2) | 20.3 (4–31.7) | 90 (70–100) | 100 (50–100) | 100 (33.3–100) | 46 (36–56) | 64 (52–76) | 75 (62.5–87.5) | 100 (80–100) | 71.3 (58.8–77.5) |
| Married | 49.9 (23.1–52.7) | 13.8 (-1.9–37.7) | 95 (75–100) | 100 (50–100) | 100 (33.3–100) | 51 (35.8–56.3) | 72 (60–84) | 75 (62.5–75) | 100 (100–100) | 77.5 (58.8–77.5) |
| P value | 0.514 | 0.735 | 0.343 | 0.978 | 0.841 | 0.867 | 0.207 | 0.837 | 0.227 | 0.632 |
| **Affordability of healthcare services** | | | | | | | | | | |
| No Difficulties | 44.1 (25.7–53.9) | 23.5 (8.3–33.7) | 90 (75–100) | 100 (75–100) | 100 (66.7–100) | 50.3 (40.8–56.3) | 68 (56–76) | 75 (62.5–87.5) | 100 (80–100) | 71.3 (58.8–83.8) |
| Difficulties | 36.4 (15.3–52.1) | 17.1 (-0.7–29.5) | 85 (65–100) | 100 (25–100) | 100 (33.3–100) | 45.5 (36–51.3) | 60 (52–72) | 75 (50–87.5) | 100 (80–100) | 65 (52.5–77.5) |
| P value | **<0.001** | **<0.001** | **<0.001** | **<0.001** | **<0.001** | **<0.001** | **<0.001** | **<0.001** | **<0.001** | **<0.001** |
| **BMI categories** | | | | | | | | | | |
| Underweight | 36.4 (16.5–50.7) | 18.7 (2.3–29.9) | 85 (65–100) | 100 (50–100) | 100 (33.3–100) | 45.5 (36–51.3) | 60 (52–72) | 75 (62.5–87.5) | 100 (80–100) | 65 (52.5–77.5) |
| Normal | 41.6 (21.6–53.8) | 20.7 (4.9–32) | 90 (70–100) | 100 (50–100) | 100 (33.3–100) | 46 (36.3–56) | 64 (52–76) | 75 (62.5–87.5) | 100 (80–100) | 71.3 (58.8–83.8) |
| Overweight | 44 (25.4–53.5) | 21.3 (3.2–32.9) | 90 (70–100) | 100 (75–100) | 100 (66.7–100) | 46.3 (40.5–56.3) | 64 (56–76) | 75 (62.5–87.5) | 100 (90–100) | 71.3 (58.8–81.3) |
| Obese | 39.3 (21.9–51.3) | 18.2 (1.6–36.6) | 90 (80–95) | 100 (25–100) | 100 (33.3–100) | 46 (36–56.3) | 72 (56–84) | 62.5 (50–87.5) | 100 (80–100) | 71.3 (58.8–77.5) |
| P value | **0.007** | 0.471 | 0.053 | 0.117 | 0.270 | **0.005** | **0.001** | 0.719 | 0.054 | **<0.001** |
| **Symptoms of COVID-19** | | | | | | | | | | |
| No symptoms | 43.2 (22.6–53.9) | 21 (5.8–32.2) | 90 (70–100) | 100 (50–100) | 100 (33.3–100) | 46 (36.3–56.3) | 64 (52–76) | 75 (62.5–87.5) | 100 (80–100) | 71.3 (58.8–81.3) |
| Atypical symptoms | 23.1 (6.1–42.1) | 10.9 (-7–27) | 85 (60–95) | 75 (25–100) | 66.7 (0–100) | 40.8 (36–46.3) | 56 (48–68) | 62.5 (50–75) | 80 (60–100) | 58.8 (46.3–71.3) |
| Typical symptoms | 35.3 (17.2–49) | 17.6 (-1–30.9) | 90 (70–95) | 75 (25–100) | 100 (33.3–100) | 45.8 (36–55.5) | 64 (52–76) | 75 (62.5–87.5) | 100 (80–100) | 68.1 (58.8–77.5) |
| P value | **<0.001** | **<0.001** | **<0.001** | **<0.001** | **<0.001** | **<0.001** | **<0.001** | **<0.001** | **<0.001** | **<0.001** |
| **Having chronic disease** | | | | | | | | | | |
| No | 40.7 (20.9–53.2) | 20.7 (5.1–31.9) | 90 (70–100) | 100 (50–100) | 100 (33.3–100) | 46 (36–56) | 64 (52–76) | 75 (62.5–87.5) | 100 (80–100) | 71.3 (58.8–81.3) |
| Yes | 40.3 (19.6–51) | 14.5 (-4.5–27.8) | 90 (70–100) | 100 (50–100) | 66.7 (0–100) | 46 (36–51.5) | 64 (52–72) | 75 (62.5–87.5) | 100 (70–100) | 65 (52.5–77.5) |
| P value | 0.457 | **0.007** | 0.621 | 0.257 | **0.001** | 0.111 | 0.312 | 0.484 | **0.021** | **0.011** |
| **FCV-19S score categories** | | | | | | | | | | |

(*Continued*)

**Table 2.** (Continued)

| | Physical Composite Score (PCS) | Mental Composite Score (MCS) | Physical functioning (PF) | Role limitations due to physical health (RP) | Role limitations due to emotional problems (RE) | Vitality (VT) | Emotional well-being (MH) | Social functioning (SF) | Pain (BP) | General health (GH) |
|---|---|---|---|---|---|---|---|---|---|---|
| Low | 43.6 (23.5–53.9) | 21.2 (3.9–32.9) | 90 (75–100) | 100 (50–100) | 100 (33.3–100) | 46.3 (40.8–56.3) | 68 (56–76) | 75 (62.5–87.5) | 100 (80–100) | 71.3 (58.8–83.8) |
| High | 31.7 (11.2–49.8) | 18.6 (3.8–29) | 85 (55–100) | 100 (25–100) | 100 (33.3–100) | 41 (36–51.3) | 56 (52–68) | 75 (62.5–87.5) | 100 (80–100) | 65 (52.5–77.5) |
| P value | **<0.001** | **<0.001** | **<0.001** | **0.023** | 0.663 | **<0.001** | **<0.001** | **0.001** | **<0.001** | **<0.001** |
| **How have your quality of life changed compare to before the pandemic** | | | | | | | | | | |
| Not Worsen | 41.9 (22.5–53.3) | 21.5 (7.4–32.5) | 90 (70–100) | 100 (50–100) | 100 (66.7–100) | 46 (36.3–56.3) | 64 (52–76) | 75 (62.5–87.5) | 100 (80–100) | 71.3 (58.8–81.3) |
| Worsen | 23.9 (3.5–45.4) | -3.5 (-14.3–17.7) | 85 (55–95) | 37.5 (0–100) | 33.3 (0–66.7) | 36.3 (31.3–46) | 58 (48–70) | 62.5 (50–75) | 90 (70–100) | 58.8 (52.5–71.3) |
| P value | **<0.001** | **<0.001** | **<0.001** | **<0.001** | **<0.001** | **<0.001** | **<0.001** | **<0.001** | **<0.001** | **<0.001** |

All data was presented as median (IQI). Statistical comparison using:

Wilcoxon rank-sum test for continuous-skewed variable—display as median(iqr) for comparison between 2 groups

Kruskal-Wallis test for continuous-skewed variables—display as median (iqr) for comparison between 3 or more groups

Bold p-value indicated statistical significance (p<0.05).

*Fifth and sixth year students are only from General Medicine and Preventive Medicine major

-11.52; 95% CI: -16 to -7). An identical pattern of association was also found in our quantile regression analysis with same dependent and independent variables (**S5 Table in S1 File**), with the only difference being that Preventive Medicine major was not statistically significant associated with lower PCS score (Coef: -3.00; 95% CI: -6.8 to 0.79).

In **Table 4**, there was also no difference in MCS scores between gender. Significant fear of COVID-19 and being on clinical rotation had no statistically significant effect on MCS score. Regarding academic characteristics, majoring in Preventive Medicine result in higher MCS score (Coef: 2.97; 95% CI: 0.33 to 5.6) compared to those who were majoring in general medicine. Lower MCS score was associated with: having chronic disease (Coef: -9.5; 95% CI: -1.7 to -5.6), having atypical symptoms of COVID-19 (Coef: -7.59; 95% CI: -12 to -3.4), difficulties affording healthcare services (Coef: -5.20; 95% CI: -7.2 to -3.2). The association of having atypical symptoms and affordability of healthcare services with MCS score was also consistent with our quantile regression analysis. However, in the quantile regression analysis with the same variables (**S6 Table in S1 File**), there were no association between the Preventive Medicine major (p = 0.064), chronic disease and MCS (p = 0.073), but being on clinical rotation was associated with a higher MCS score (Coef: 3.239; 95% CI: 0.71 to 5.8).

**Perceived negative changes in quality of life.** Overall, 9.9% (95% CI: 8.5%– 11.4%) of the study population reported a significant negative change in quality of life. 20.2% of 6th-year medical students reported having a reduction in quality of life, a prevalence much higher than that of their junior (<10%) (p<0.01). Also, students that were on clinical rotation reported a negative change in quality of life was higher (11.4%) compared to the who weren't (6.9%) (p<0.01). A significant portion of those who were obese (28%) or having chronic disease (20.9%) perceived a decrease in the quality of life, while this decrease is much less prevalent in students with normal BMI (9.8%) or those without chronic disease (9.0%) (p<0.05). There

**Table 3. Linear regression of PCS score.**

| | Physical Composite Score | | |
| --- | --- | --- | --- |
| | Coef. | p-value | 95% CI | |
| **Academic Major** | | | | |
| General Medicine | ref. | | | |
| Preventive Medicine | -3.31 | **0.024** | -6.2 | -0.43 |
| Nursing | -5.67 | **0.001** | -9 | -2.4 |
| **Gender** | | | | |
| Female | ref. | | | |
| Male | -0.62 | 0.629 | -3.1 | 1.9 |
| **Currently on Clinical Rotation** | | | | |
| No | ref. | | | |
| Yes | 8.89 | **<0.001** | 6.6 | 11 |
| **Marital Status** | | | | |
| Single | ref. | | | |
| Married | 3.35 | 0.612 | -9.6 | 16 |
| **Affordabilities of Healthcare Services** | | | | |
| No Difficulties | ref. | | | |
| Difficulties | -6.32 | **<0.001** | -8.5 | -4.1 |
| **BMI Category** | | | | |
| Underweight | ref. | | | |
| Normal | 2.28 | 0.108 | -0.5 | 5.1 |
| Overweight | 3.44 | 0.095 | -0.59 | 7.5 |
| Obese | -1 | 0.828 | -10 | 8 |
| **Having Chronic Disease** | | | | |
| No | ref. | | | |
| Yes | -0.35 | 0.868 | -4.5 | 3.8 |
| **Symptoms of COVID-19** | | | | |
| No symptoms | ref. | | | |
| Typical Symptoms | -4.96 | **0.003** | -8.2 | -1.7 |
| Atypical Symptoms | -11.52 | **<0.001** | -16 | -7 |
| **Fear of COVID-19 Scale score categories** | | | | |
| Low | ref. | | | |
| High | -5.39 | **<0.001** | -7.8 | -3 |

were no significant differences among genders (p = 0.25), marital status (p = 0.35), academic majors, and FCV-19S score (p = 0.08). **(S7 Table in S1 File)**

Table 5 shows the modified Poisson regression analysis of perceived negative changes in the quality of life of medical students. Results showed that being on clinical rotation was associated with more deterioration in quality of life during COVID-19 (PR: 1.5; 95% CI: 1.05–2.2). Additionally, students that had difficulties affording health services were more susceptible to these negative changes compared to those who don't (PR: 1.4; 95% CI: 1.02–1.95). Obesity (PR: 2.38; 95% CI: 1.08–5.25), having chronic disease (PR: 1.92; 95% CI: 1.23–3), and typical symptoms (PR: 1.85; 95% CI: 1.23–2.78), atypical symptoms of COVID-19 (PR: 2.32; 95% CI: 1.41–3.81) were also associated with the reduction in quality of life of the sample.

## Discussion

From our study, it can be seen that there is a similar pattern in quality of life between different groups of students: a lower score in vitality (VIT = 46), emotional functioning (MH = 64) and

**Table 4. Linear regression of MCS score.**

| | Mental Composite Score | | | |
|---|---|---|---|---|
| | Coef. | p-value | 95% | CI |
| **Academic Major** | | | | |
| General Medicine | ref. | | | |
| Preventive Medicine | 2.99 | **0.028** | 0.33 | 5.6 |
| Nursing | 0.34 | 0.827 | -2.7 | 3.4 |
| **Gender** | | | | |
| Female | ref. | | | |
| Male | 0.53 | 0.651 | -1.8 | 2.9 |
| **Currently on Clinical Rotation** | | | | |
| No | ref. | | | |
| Yes | 1.29 | 0.238 | -0.85 | 3.4 |
| **Marital Status** | | | | |
| Single | ref. | | | |
| Married | -1.06 | 0.862 | -13 | 11 |
| **Affordabilities of Healthcare Services** | | | | |
| No Difficulties | ref. | | | |
| Difficulties | -5.2 | **<0.001** | -7.2 | -3.2 |
| **BMI Category** | | | | |
| Underweight | ref. | | | |
| Normal | 0.64 | 0.622 | -1.9 | 3.2 |
| Overweight | 1.1 | 0.563 | -2.6 | 4.8 |
| Obese | 1.71 | 0.687 | -6.6 | 10 |
| **Having Chronic Disease** | | | | |
| No | ref. | | | |
| Yes | -5.6 | **0.004** | -9.5 | -1.7 |
| **Symptoms of COVID-19** | | | | |
| No symptoms | ref. | | | |
| Typical Symptoms | -2.38 | 0.125 | -5.4 | 0.66 |
| Atypical Symptoms | -7.59 | **<0.001** | -12 | -3.4 |
| **Fear of COVID-19 Scale score categories** | | | | |
| Low | ref. | | | |
| High | -0.95 | 0.4 | -3.2 | 1.3 |

social functioning (SF = 75) with vitality being the lowest. This pattern is similar to a pattern from other studies on medical students using the SF-36 instruments, outside of the pandemic [51, 52]. A study on Italian medical students from 2005–2015 showed a similar pattern of lower VIT (Mean: 59.4; SD: 16.1), MH (Mean: 68.65; SD: 16.3) compared to other dimensions and reference Italian population [52]. Compared to the general population, medical students were already suffering from burnout, stress and multiple mental issues, as mentioned in previous studies in other countries [15–18]. Combined with sudden changes in the context of the pandemic, students are having symptoms of depression, anxiety, sleep disturbances [22, 23, 25, 53]. Low vitality, fatigue, extensive feelings of sleepiness and increased daily nap time were recorded in multiple studies [53–55].

An extremely low level of mental quality of life was recorded as 98.2% of students have a lower MCS score (Median: 20.3 IQR: 3.8–31.7) than that of the reference population (MCS: 50) The extreme number could be partially explained due to the reference Vietnamese population being young with an average age of 27 and had a high quality of life, and the below-

**Table 5. Poisson regression analysis of negative changes in quality of life.**

| | Changes in Quality of Life | | | |
|---|---|---|---|---|
| | PR | p-value | 95% | CI |
| **Academic Major** | | | | |
| General Medicine | ref. | | | |
| Preventive Medicine | 1.10 | 0.631 | 0.74 | 1.65 |
| Nursing | 0.92 | 0.742 | 0.55 | 1.53 |
| **Gender** | | | | |
| Female | ref. | | | |
| Male | 1.22 | 0.282 | 0.85 | 1.76 |
| **Currently on Clinical Rotation** | | | | |
| No | ref. | | | |
| Yes | 1.52 | **0.028** | 1.05 | 2.22 |
| **Marital Status** | | | | |
| Single | ref. | | | |
| Married | 1.57 | 0.529 | 0.39 | 6.42 |
| **Affordabilities of Healthcare Services** | | | | |
| No Difficulties | ref. | | | |
| Difficulties | 1.41 | **0.037** | 1.02 | 1.95 |
| **BMI Category** | | | | |
| Underweight | 0.82 | 0.371 | 0.54 | 1.26 |
| Normal | ref. | | | |
| Overweight | 0.85 | 0.522 | 0.52 | 1.40 |
| Obese | 2.38 | **0.032** | 1.08 | 5.25 |
| **Having Chronic Disease** | | | | |
| No | ref. | | | |
| Yes | 1.92 | **0.004** | 1.23 | 3.00 |
| **Symptoms of COVID-19** | | | | |
| No symptoms | ref. | | | |
| Typical Symptoms | 1.85 | **0.003** | 1.23 | 2.78 |
| Atypical Symptoms | 2.32 | **0.001** | 1.41 | 3.81 |
| **FCV-19S score categories** | | | | |
| Low | ref. | | | |
| High | 0.71 | 0.078 | 0.49 | 1.04 |

average medical students quality of life suggested in multiple studies outside of the pandemic [51, 52]. However, the median MCS score was still exceptionally low and thus, the overall mental status of the students should be critically alarming. In our study, mental quality of life was found to be associated with academic major (Coef: 2.97; 95% CI: 0.33 to 5.6), affordability of healthcare services (Coef: -5.20; 95% CI: -7.2 to -3.2), chronic disease (Coef: -9.5; 95% CI: -1.7 to -5.6), and atypical symptoms of COVID-19 (Coef: -7.59; 95% CI: -12 to -3.4), while not being associated with gender. The association of financial difficulties and quality of life is rather similar to the findings of a study on depression on 4th, 5th and 6th-year students of Hanoi Medical University before the pandemic, in 2019 [18]. However, a study by Vo on medical students in southern Vietnam, in the same year 2019, shows that female has a lower mental and physical quality of life [56]. This variation in results could be explained by the difference in environmental exposure between the two cohorts as there are major differences in the curriculum and academic environments, as some southern medical schools used an organ-based and modular curriculum for general medicine program, while Hanoi Medical University still

used a traditional discipline-based curriculum for general medicine program at the time of this survey.

Regarding physical health, the PCS score of the sample has a median score of 40.6 with 66.5% below the population average. The low level of physical health is likely to be related to the low level of mental health, vitality and increased fatigue reduced physical activities during the lockdown [55, 57–59]. There have been mixed findings regarding gender and physical-related quality of life around the world and Vietnam [56, 60, 61]. Similar to mental quality of life., physical quality of life was lower in females in a study on southern medical students of Vietnam [56]. However, this association was also not observed in our study. On the other hand, difficulties affording healthcare services (Coef: -6.32; 95% C: -8.5 to -4.1), fear of COVID-19 (Coef: -5.39; 95% CI: -3 to -7.8), and symptoms of COVID-19: typical (Coef: -4.96; 95% CI: -8.2 to -1.7) and atypical (Coef: -11.52; 95% CI: -16 to -7) were found to be correlated with reduced physical health. In addition, academic major and clinical rotation also associated with physical health, as preventive medicine (Coef: -3.3; 95% CI: -6.2 to -0.43) and nursing students (Coef: -5.67; 95% CI: -9 to -2.4) have lower PCS scores and students being on clinical rotation have better physical health (Coef: 8.89; 95% CI: 6.6 to 11).

There was no relationship between genders and reduction in quality of life in our study, consistent with findings on PCS and MCS scores. Significant perceived negative changes in quality of life were associated with being on clinical rotation (PR: 1.5; 95% CI: 1.05–2.2), difficulties affording healthcare services (PR: 1.41; 95% CI: 1.02–1.95), obesity (PR: 2.38; 95% CI: 1.08–5.25), having chronic diseases (PR: 1.92; 95% CI: 1.23–3), and having symptoms of COVID-19: typical symptoms (PR: 1.85; 95% CI: 1.23–2.78), atypical symptoms of COVID-19 (PR: 2.32; 95% CI: 1.41–3.81). Despite having higher PCS and relatively the same MCS score, students on clinical rotation are more likely to perceive negative changes in their quality of life (PR: 1.5; 95% CI: 1.05–2.2), which is probably due to visible disruption in clinical rotation and medical rotation and concerns for their career development, which was explored in prior studies [23, 25]. Students who have difficulties affording healthcare services are 1.41 times (PR: 1.41; 95% CI: 1.02–1.95) more likely to feel that their quality of life has worsened. Affordability of healthcare services consistently correlated with changes in quality of life and quality of life of students during the pandemic, as it could place an invisible pressure on the person should they get infected. According to our study, students who have obesity and/or chronic disease are more vulnerable to deterioration in their quality of life. Compared to a person with a normal BMI, an obese person is 2.38 times (PR: 2.38; 95% CI: 1.08–5.25) more likely to have worse quality of life in the pandemic, while having chronic diseases increases the prevalence 1.92 times (PR: 1.92; 95% CI: 1.23–3). An interesting finding of our study is that atypical symptoms had an impact on MCS scores (Coef: -7.59; 95% CI: -12 to -3.4) while typical symptoms did not. The impact of atypical symptoms on mental quality of life could be due to the physical impairment it brings and the uncertainty in making the decision to get a COVID-19 test [62]. However, there was no association between Fear of COVID-19 and reduction in quality of life (PR: 0.71; 95% CI: 0.49–1.04). Therefore, we hypothesized that the symptoms of COVID-19 mostly affected medical students via physical impairment, rather than fear and anxiety because Fear of COVID-19 was not found to be associated with mental quality of life but had some relation to the physical quality of life. This result could be explained by the restrictions of physical activities during the lockdown, while the pandemic was relatively controlled. Therefore, Fear of COVID-19 could have a lesser impact than expected on medical students' mental health and anxiety during the pandemic, as the change was primarily due to previously mentioned factors.

## Strength and limitations

The strength of our study is that our representative sample frame included all students from 3 majors: General Medicine, Preventive Medicine, and Nursing. Moreover, the main instrument —the Vietnamese version of the SF-36 was validated and was shown to provide a valid assessment of self-reported health status among the Vietnamese population and allowed for both compound and specific evaluation of quality life. Finally, the survey was conducted after the first wave of COVID hit Vietnam—after Bach Mai hospital, a major teaching hospital of Hanoi Medical University, was locked down after a case series COVID-19 on March 28[th], 2020. Therefore, we believe that the result captures the changes in quality of life of the students during the early stage of the pandemic. However, this study used an online, anonymous data collection scheme, which lead to a low response rate of only about 43%, and potential sampling errors, and selection bias. The difference in response rate between academic majors could be due to clinical rotation's scheduling, the willingness and interest of the class representatives, or miscommunications between the Office for Student Services and the class representatives. Due to the anonymous feature of the survey, we were not able to pinpoint the exact reasons for the difference in response rate and will look into this issue in future studies. Moreover, there was no available data on medical students' quality of life using the same instruments prior the pandemic, so a direct comparison was not possible. The use of a cross-sectional study design also limited our interpretation of the results to association, rather than causation. This design was also unable to examine the changes in quality of life of the students in other waves of COVID-19 and their adaptation going further into the pandemic. Also, the translation into Vietnamese of the SF-36, FCV-19S might affect the validity of these questionnaire. Furthermore, the SF-36 tools scoring used population-based scoring methods, however, there was no Vietnamese population-derived weighting coefficient for 8 dimensions of SF-36 when calculating PCS and MCS score, thus we had to use the coefficient from the original USA population, which can cause a slight alteration in the results, as the weight for each population is slightly different [44]. Additionally, since we only use a single direct question to examine the reduction in quality of life, it is difficult to accurately determine the specific changes in quality of life of healthcare students in the pandemic.

## Conclusion

The physical and mental quality of life of the medical and nursing students was lower than that of the general population. Vitality, mental health, and social functioning dimensions showed the lowest score among the 8 dimensions. Lower PCS score was associated with academic majors, chronic disease, not being on clinical rotation, difficulties affording healthcare services, and Fear of COVID-19. Lower MCS score was associated with academic major, difficulties affording healthcare services, and chronic disease. Perceived reduction in quality of life was associated with being on clinical rotation, difficulties affording healthcare services, having chronic disease, obesity, and having symptoms of COVID-19. Future studies could be done to explore further the specific impact of COVID-19 on students of different major in regard to their curriculum, risk, and exposure in the pandemic.

## Supporting information

**S1 File.**
(DOCX)

## Author Contributions

**Conceptualization:** Le Dai Minh, Hoang Huy Phan, Duong Ngoc Le Mai, Nguyen Tien Dat, Ngo Minh Tri, Nguyen Viet Ha, Nguyen Huu Tu, Kirsty Foster, Kim Bao Giang, Tung Thanh Pham.

**Data curation:** Le Dai Minh, Duong Ngoc Le Mai, Nguyen Huu Tu, Kim Bao Giang.

**Formal analysis:** Le Dai Minh, Hoang Huy Phan, Duong Ngoc Le Mai, Tung Thanh Pham.

**Investigation:** Le Dai Minh, Hoang Huy Phan, Kim Bao Giang, Tung Thanh Pham.

**Methodology:** Le Dai Minh, Hoang Huy Phan, Duong Ngoc Le Mai, Kim Bao Giang, Tung Thanh Pham.

**Project administration:** Le Dai Minh, Kim Bao Giang, Tung Thanh Pham.

**Resources:** Tung Thanh Pham.

**Writing – original draft:** Le Dai Minh, Hoang Huy Phan, Duong Ngoc Le Mai, Nguyen Tien Dat, Ngo Minh Tri, Nguyen Viet Ha, Nguyen Huu Tu, Kirsty Foster, Kim Bao Giang, Tung Thanh Pham.

**Writing – review & editing:** Le Dai Minh, Hoang Huy Phan, Duong Ngoc Le Mai, Nguyen Tien Dat, Ngo Minh Tri, Nguyen Viet Ha, Nguyen Huu Tu, Kirsty Foster, Kim Bao Giang, Tung Thanh Pham.

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
