## [Decision Letter · Decision Letter 0]

8 Jul 2022

PONE-D-22-14424Pattern and perceived changes in quality of life of Vietnamese medical and nursing students during the COVID-19 pandemicPLOS ONE

Dear Dr. Pham,

Thank you for submitting your manuscript to PLOS ONE. After careful consideration, we feel that it has merit but does not fully meet PLOS ONE’s publication criteria as it currently stands. Therefore, we invite you to submit a revised version of the manuscript that addresses the points raised during the review process. Three reviewers have engaged with your study. All the reviewers see your study relevant and with the potential for publication subject to some revisions. Please critically engage with the reviewer comments and resubmit for reconsideration.

We look forward to receiving your revised manuscript.

Kind regards,

Christmal Dela Christmals, PhD, MSc, BSc, RN

Academic Editor

PLOS ONE

Journal Requirements:

4. We note you have included a table to which you do not refer in the text of your manuscript. Please ensure that you refer to Table 3 an 4 in your text; if accepted, production will need this reference to link the reader to the Table.

Reviewers' comments:

Reviewer's Responses to Questions

**Comments to the Author**

1. Is the manuscript technically sound, and do the data support the conclusions?

Reviewer #1: Yes

Reviewer #2: Yes

Reviewer #3: Partly

2. Has the statistical analysis been performed appropriately and rigorously? 

Reviewer #1: Yes

Reviewer #2: Yes

Reviewer #3: Yes

3. Have the authors made all data underlying the findings in their manuscript fully available?

Reviewer #1: Yes

Reviewer #2: Yes

Reviewer #3: No

4. Is the manuscript presented in an intelligible fashion and written in standard English?

Reviewer #1: Yes

Reviewer #2: Yes

Reviewer #3: No

5. Review Comments to the Author

Reviewer #1: PONE D 22 14424

COMMENTS

The manuscript is technically sound and the data support all the conclusions made

L26- L 61 Credentials – some authors do not have Orcid numbers. It will be beneficial to generate an Orcid number

L 64 ABSTRACT- abstract is concise and succinct.

L67 add nursing to medical students

L 101 add s to student

Introduction – well explained and purpose of the study well explained

Methods

The test and sample size are appropriate

The language used is clear, correct, and unambiguous

Setting – well explained

Survey instrument – questionnaire items well explained

Sample size

L221 - remove full stop after percentage of nursing students

L223 – Replace “an” with a in “A self-reported…”

L225 – Who are these representatives?

Data analysis

L235 – Separate the ap-value to read a p-value. Data analysis procedure explained well

Ethical issues are well explained by authors

Results

Results well explained

Table concise

L291 – What does normal students mean? Try another word to describe them as this does not sound good

Discussion

Discussion covered the main findings in the study and exposed challenges face by health students on clinical rotation during the Covid -19 pandemic.

Strengths and limitations

Strengths and limitations well explained in the study

Conclusion

Replace sample on L471 with group names of participants

Reviewer #2: The manuscript is well written, with few grammatic errors. See attached document with comments.

Reviewer #3: The experiments and the statistical analysis seem to have been conducted rigorously. Data underlying the findings is not made fully available. The authors in the manuscript indicated the following in this regards: "According to our application to I have no competing interests, the data cannot be shared publicly because of ethical restrictions to protect the confidentiality of the participants. A de-identified dataset is available for researchers who meet the criteria for access to confidential data." The reviewer understands and respect that the Institutional Review Board of Hanoi University of Public Health wants to protect the confidentiality of participants in this way. The reviewer in this regard wants to suggest that the authors should consider omitting the name of the University to not only further protect the confidentiality of the participants but also of the University.

In rule 136-138 the authors mentioned that the cross-sectional study was conducted in the first six months of the COVID-19 pandemic and then go further to state the dates on which the study was conducted namely 7-29 April of 2020. The sentence could cause confusion as it could sound as if the study was conducted over a period of six months.

In the discussion of the limitations of the study, the authors seem to contradict themselves with regards to utilizing a translated version of the SF-36 and of the FCV-19S. The authors first mentioned that the main instrument of the study namely the Vietnamese version of the SF-36 was validated and as a result provided a valid assessment of self-reported health status among the Vietnamese population. It further allowed for both compound and specific evaluation of quality life. Then in the discussion of the limitation of the study, the authors indicated that the translation of the SF-36 and the FCV-19S into Vietnamese might have affected the validity of these questionnaires.

The reviewer wonders whether the research was not conducted too hastily which resulted in preventing the authors to conduct a pilot study beforehand in order to rule out the possible limitations that the translation of the documents in the end brought about.

The reviewer wants to suggest that if the manuscript is to be published that the authors add information pertaining to what was put into place to support participants who, as a result of the study, needed psychological treatment/assistance. From the discussion of the results it is for instance mentioned that the quality of life for some of the participants changed as a result of the COVID-19 pandemic. By participating in the study could have created a specific awareness amongst some of the participants about these changes which potentially could for instance have contributed to further stress and anxious feelings. The authors in Rules 374-375 for instance refer to what the mental state of medical students could be as a result of the sudden changes in the context of the pandemic.

The reviewer is of the opinion that the results of the specific topic lends itself to causation interpretation. The authors stated that the use of a cross-sectional study design however limited their interpretation of the results to association, rather than causation. The interpretation of the conclusions therefore seems to be very linearly done. The authors do not draw adequate connections between the outcome of the study and the quality of life of the different groups that participated.

The formulation and construction of sentence for instance need attention (Refer to Rules 196 and 197: Should the formulation of the last part of the sentence not read: and any score below 50 was considered to be "Below the population average". etc). In certain sentences words such as ‘the’ and ‘a’ are missing (Refer to Rule 171: The word "the" should be added between the words "to" and WHO"s etc). COVID-19 is sometimes indicated as Covid-19 (Refer to Rules 101, 137, 156, 174 etc). Other examples where attention needs to be paid to the editing of the manuscript involves the following: Rule 3: it should be students and not just "student" as it currently stands; Rule 136: the "who" needs to be added between the words "student" and "major".

It is strongly suggested that the manuscript should be language edited if it is to be reviewed again.

6. PLOS authors have the option to publish the peer review history of their article (what does this mean?). If published, this will include your full peer review and any attached files.

Reviewer #1: No

Reviewer #2: No

Reviewer #3: No

---

## [Author Response · Author response to Decision Letter 0]

25 Aug 2022

Journal Requirements:

Thank you for your comments, we checked and revised the manuscript to meet the PLOS ONE's style requirements.

Thank you for your comments, all participants received comprehensive information about the survey and so were fully informed before giving consent (by clicking “I agree to participate” button on the informed consent page of the questionnaire). We added this information on the “Ethical issue” section as you requested (Line 251)

According to our application to the Institutional Review Board (IRB) of Hanoi University of Public Health, the data cannot be shared publicly because of ethical restrictions to protect the confidentiality of the participants imposed by the IRB. A de-identified dataset is available for researchers who meet the criteria for access to confidential data. Requests for data should be submitted to Institutional Review Board of Hanoi University of Public Health

(irb@huph.edu.vn) and the corresponding author, Dr. Pham Thanh Tung (phamthanhtung@hmu.edu.vn)

4. We note you have included a table to which you do not refer in the text of your manuscript. Please ensure that you refer to Table 3 an 4 in your text; if accepted, production will need this reference to link the reader to the Table.

Reviewers' comments:

Reviewer's Responses to Questions

Comments to the Author

Reviewer 1 Comments

Thank you for your very insightful comments. These comments greatly helped us complete and improve our manuscript. We will be replying to every specific comment down below.

Comment: 

The manuscript is technically sound and the data support all the conclusions made

L26- L 61 Credentials – some authors do not have Orcid numbers. It will be beneficial to generate an Orcid number

L 64 ABSTRACT- abstract is concise and succinct.

Answer: Thank you for your comments. We will ask all of our co-authors to include this information

Comment: L67 add nursing to medical students

Answer: we have fixed the sentence as recommended in the manuscript

Comment: L 101 add s to student

Answer: we have added “s” to student in the keyword section as recommended

Comment: 

Introduction – well explained and purpose of the study well explained

Methods

The test and sample size are appropriate

The language used is clear, correct, and unambiguous

Setting – well explained

Survey instrument – questionnaire items well explained

Answer: Thank you for your comments

Sample size

Comment: L221 - remove full stop after percentage of nursing students

Answer: Thank you for your comments. We have removed the full stop in the sentence

Comment: L223 – Replace “an” with a in “A self-reported…”

Answer: Thank you for your comments. We have replaced the “an” with “a” in the sentence.

Comment: L225 – Who are these representatives?

Answer: Thank you for your comments. We added the following information on line 228

“The class representatives are students that are responsible for major communication between the school administration and the student of the class that they’re in. There is one class representative for each class, and this practice is common for most universities in Vietnam”

Data analysis

Comment: L235 – Separate the ap-value to read a p-value. Data analysis procedure explained well

Answer: Thank you for your comments. we have fixed the sentence as recommended.

Comment: 

Ethical issues are well explained by authors

Results

Results well explained

Table concise

Answer: Thank you for your comments

Comment: L291 – What does normal students mean? Try another word to describe them as this does not sound good

Answer: Thank you for your comments. We have rephrased the sentence as below on line 297

“The PCS score was significantly lower in those who had symptoms of COVID-19 (p <0.001) and those who have significant fear of COVID-19 (p <0.001), compared to those who didn’t.”

Comment:

Discussion

Discussion covered the main findings in the study and exposed challenges face by health students on clinical rotation during the Covid -19 pandemic.

Strengths and limitations

Strengths and limitations well explained in the study

Answer: Thank you for your comments

Conclusion

Comment: Replace sample on L471 with group names of participants

Answer: Thank you for your comments. We have rephrased the sentence as below on line 477

“The physical and mental quality of life of the medical and nursing students was lower than that of the general population”

 

Reviewer 2 Comments

Thank you for your very insightful comments. These comments greatly helped us complete and improve our manuscript. We will be replying to every specific comment down below.

Comment: How many is all?

Answer: Thank you for your comments. The total number of participants was 3672 students. We revised line 75-76 as follows:

“The study was a cross-sectional study on all students of Hanoi Medical University from 3 majors: General Medicine, Preventive Medicine, Nursing (3672 invited students)”

Comment: The response rate seems low

Answer: Thank you for your comments. Due to using online questionnaires to collect data, the response rate is usually low. This was acknowledged in our limitation section in the full text on line 459:

“However, this study used an online, anonymous data collection scheme, which lead to a low response rate of only about 43%, and potential sampling errors, and selection bias.”

Comment: Would be good to highlight, what leads to the previous mental problem faced by the students for the reader to follow the progression of the risk factors. 

Answer: Thank you for your comments. We revised it as follows on line 116

“Medical and nursing students, before the pandemic, were already known to face multiple physical and mental problems including burnout, anxiety, depression, and other mental health issues, with stress, lack of academic motivation, and financial hardship being important risk factors [13–16]”

Comment: Please include literature, that address what in quality life in a life of Medical and Nursing student, and how it affects them so as to link the following sentence that address the solution which is the study aim.

Answer: Thank you for your comments. This sentence was intended as our hypothesis rather than information from the literature. We revised it as follows on line 126

“Together with other socio-economic aspects related to the pandemic like lockdown, social distancing, limitation in physical activities, the authors hypothesized that the physical and mental quality of life of medical students could be at high risk and can potentially lead to poorer health outcomes.”

Comment: The authors, instead of we.

Answer: Thank you for your comments. We revised the sentence as you suggested

Comment: Would be good to add the overall medical and nursing students that were enrolled in that period for the reader to see the sample size you got for the study.

Answer: Thank you for your comments. We revised the sentence as you suggested on line 138.

“A cross-sectional study was conducted on 1583 students who major in medicine, preventive medicine, and nursing, at Hanoi Medical University (out of total 3672 students), from 7th to 29th of April 2020. This period is within the first 6 months of the covid-19 pandemic in Vietnam”

Comment: Ensure consistency. Second or 2nd ?

Answer: Thank you for your comments. We have fixed the typing error (from 2nd to second) to ensure consistency

Comment: Is the instrument developed, adapted or edited by the authors and is the permission ranted if it was adapted ?

Answer: Thank you for your comments. We included detailed information on these questionnaires in our method section. No adaptation or edition was made by the authors as we simply used the Vietnamese version (adapted, translated, tested, and sometimes validated in previous studies)

Line 176 “The FCV-19S questionnaire was translated into Vietnamese in a previous study and showed good item-scale convergent validity (mean of Rho = 0.77), discriminant validity and construct validity and high internal consistency (Cronbach’s alpha = 0.90) with the Vietnamese translation [33,34]”. The information on this questionnaire was available online from previous paper and could be considered resource on public domain, which could be reused by other researchers.

Line 187 “The quality of life of students during the period was determined using the Vietnamese version of the Quality-of-life SF-36 version 2.0 [35]”. The information on this questionnaire was available online from previous paper and could be considered resource on public domain, which could be reused by other researchers.

The license for this questionnaire was also fully granted by RAND: https://www.rand.org/health-care/surveys_tools/mos/36-item-short-form/terms.html

Comment: Acknowledge the source, since it is adapted please

Answer: Thank you for your comments. We included the name of the author of the article in the source and cited accordingly on line 177

“Fear of Covid-19 was assessed using the FCV-19S questionnaire. The FCV-19S questionnaire was translated into Vietnamese in a previous study by Nguyen and colleague and showed good item-scale convergent validity (mean of Rho = 0.77), discriminant validity and construct validity and high internal consistency (Cronbach’s alpha = 0.90) with the Vietnamese translation [33,34].”

Comment: Suggested a usage of nor instead of or

Answer: Thank you for your comments. We have changed it to “neutral” as in the original English version

https://www.nlm.nih.gov/dr2/Fear_of_Covid-19_Scale_2020.pdf

Line 180: “It consisted of 7 items and utilized a 5-point Likert scale with 1 = “strongly disagree”, 2 = “disagree”, 3 = “neutral”, 4 = “agree”, 5 = “strongly agree””

Comment: What informed then number of invited participants. It needs be scientifically justifiable

Answer: Thank you for your comments. The total number of invited students was collected using record from the administration. As for the scientifical justification, we invited ALL students in Doctor of General Medicine, Doctor of Preventive Medicine, and Bachelor of Nursing tracks (3672 students) rather than using random sampling. Inviting all individuals from the target population to participate is always the best sampling method if resources are available.

We also provided information on sample size calculations on line 211:

“Sample size and data collection

The required sample size was estimated by using the formula for estimating sample mean [38], as follows:

 n=(Z_(1-α/2)^2 〖*σ〗^2)/d^2 =(〖1.96〗^2*〖12.5〗^2)/1^2 =600.25

With, n being the required sample size, Z being the standard error associated with the chosen level of confidence 5%, σ being the standard deviation of the general quality of life score taken from the population used to validate the Vietnamese version of the SF-36 [35].”

Comment: Clarify the ap to the reader

Answer: Thank you for your comments. It is a typo, we changed it to:

Line 241: “We consider a p -value < 0.05 as statistically significant for all statistical tests”

Comment: Add a source to reference this, it is significant but elaborate as to according to who

Answer: Thank you for your comments. We included citation on paper tracking and discussing the origin of this threshold. We understand this threshold is not perfect; therefore, we always try to provide full p-value rather than using cut-off and also use 95%CI when possible.

Comment: Provide value to the reader

Answer: Thank you for your comments. We used general medicine was used as a reference in the regression model for the independent variable of major, so the coefficient for general medicine was 0 while the coefficient for nursing was not significantly different from the reference, but the table did include this information. The coefficient could be understood as the difference among groups when comparing the score.

We also added detailed values in the following sentence on line 337:

“Regarding academic characteristics, majoring in Preventive Medicine result in higher MCS score (Coef: 2.97; 95% CI: 0.33 to 5.6) compared to those who were majoring in general medicine. Lower MCS score was associated with: having chronic disease (Coef: -9.5; 95% CI: -1.7 to -5.6), having atypical symptoms of COVID-19 (Coef: -7.59; 95% CI: -12 to -3.4), difficulties affording healthcare services (Coef: -5.20; 95% CI: -7.2 to -3.2). The association of having atypical symptoms and affordability of healthcare services with MCS score was also consistent with our quantile regression analysis. However, in the quantile regression analysis with the same variables (S6 Table), there were no association between the Preventive Medicine major (p=0.064), chronic disease and MCS (p=0.073), but being on clinical rotation was associated with a higher MCS score (Coef: 3.239; 95% CI: 0.71 to 5.8).”

Comment: Delete douplicate

Answer: Thank you for your comments we deleted the duplicate

Comment: This affirms the earlier suggestion, that if the tool is amended. Its reliability and validity must be ensured in the current context

Answer: Thank you for your comments. All questionnaires in languages other than English usually involve translation. Although these questionnaires may be validated rigorously. The reliability and validity will never be 100% as the original version in the original language. Even a sensitivity and specificity of 99% is not 100%. Therefore, in this sentence, we want to clearly state this fact even though previous studies may have already validated these tools rigorously in Vietnamese population. 

Reviewer 3 Comments

Thank you for your very insightful comments. These comments greatly helped us complete and improve our manuscript. We will be replying to every specific comment down below.

Comment: The experiments and the statistical analysis seem to have been conducted rigorously. Data underlying the findings is not made fully available. The authors in the manuscript indicated the following in this regards: "According to our application to I have no competing interests, the data cannot be shared publicly because of ethical restrictions to protect the confidentiality of the participants. A de-identified dataset is available for researchers who meet the criteria for access to confidential data." The reviewer understands and respect that the Institutional Review Board of Hanoi University of Public Health wants to protect the confidentiality of participants in this way. The reviewer in this regard wants to suggest that the authors should consider omitting the name of the University to not only further protect the confidentiality of the participants but also of the University.

Answer: Thank you for your comments. The dataset of this paper was collected at only one university, and the IRB did not prohibit us from reporting the name of the university from this data as there is no sensitive data regarding the university. We think that reporting group level data like table and figure are good enough to protect the confidentiality of participants 

Comment: In rule 136-138 the authors mentioned that the cross-sectional study was conducted in the first six months of the COVID-19 pandemic and then go further to state the dates on which the study was conducted namely 7-29 April of 2020. The sentence could cause confusion as it could sound as if the study was conducted over a period of six months.

Answer: Thank you for your comments. We have adjusted the sentence in the manuscript as follows, in order to clear the confusion.

Line 138: “A cross-sectional study was conducted on 1583 students major in medicine, preventive medicine, and nursing, at Hanoi Medical University (out of total 3672 students), from 7th to 29th of April 2020. This period is within the first 6 months of the covid-19 pandemic in Vietnam.”

Comment: In the discussion of the limitations of the study, the authors seem to contradict themselves with regards to utilizing a translated version of the SF-36 and of the FCV-19S. The authors first mentioned that the main instrument of the study namely the Vietnamese version of the SF-36 was validated and as a result provided a valid assessment of self-reported health status among the Vietnamese population. It further allowed for both compound and specific evaluation of quality life. Then in the discussion of the limitation of the study, the authors indicated that the translation of the SF-36 and the FCV-19S into Vietnamese might have affected the validity of these questionnaires.

The reviewer wonders whether the research was not conducted too hastily which resulted in preventing the authors to conduct a pilot study beforehand in order to rule out the possible limitations that the translation of the documents in the end brought about.

Answer: Thank you for your comments. All questionnaires in languages other than English usually involve translation. Although these questionnaires may be validated rigorously. The reliability and validity will never be 100% as the original version in the original language. Even a sensitivity and a specificity of 99% is not 100%. 

Therefore, in this sentence, we want to clearly state this limitation even though previous studies may have already validated these tools rigorously in Vietnamese population.

Comment: The reviewer wants to suggest that if the manuscript is to be published that the authors add information pertaining to what was put into place to support participants who, as a result of the study, needed psychological treatment/assistance. From the discussion of the results, it is for instance mentioned that the quality of life for some of the participants changed as a result of the COVID-19 pandemic. By participating in the study could have created a specific awareness amongst some of the participants about these changes which potentially could for instance have contributed to further stress and anxious feelings. The authors in Rules 374-375 for instance refer to what the mental state of medical students could be as a result of the sudden changes in the context of the pandemic. 

Answer: Thank you for your comments. The questionnaire was anonymous, so we could not refer student to a clinician. This anonymous procedure was put in place to protect the participants’ identity and encourage them to participate in the study as many students with mental health issue don’t want to reveal this information. The IRB also agreed and approved this anonymous procedure.

Comment: The reviewer is of the opinion that the results of the specific topic lends itself to causation interpretation. The authors stated that the use of a cross-sectional study design however limited their interpretation of the results to association, rather than causation. The interpretation of the conclusions therefore seems to be very linearly done. The authors do not draw adequate connections between the outcome of the study and the quality of life of the different groups that participated.

Answer: Thank you for your comments. The connection between quality of life and associated factors was discussed with regards to the data collected. There are many factors that need to be considered to use causation interpretation in this case. We have a low response rate, many unmeasured/residual confounders, and weak temporarily relationship in this study. All of these factors could completely change the results: student with low quality of life don’t want to participate, a strong unmeasured confounder like academic motivation distorted the relationship, the relationship is not causal as low quality of life lead to the risk factor (not the other way). 

Therefore, we are not confidence to put forward a causal interpretation and would only report association.

Comment: The formulation and construction of sentence for instance need attention (Refer to Rules 196 and 197: Should the formulation of the last part of the sentence not read: and any score below 50 was considered to be "Below the population average". etc). 

Answer: Thank you for your comments. We have rephrased the sentence accordingly.

Line 198: “A PCS/MCS score of 50 represents the reference population average, and any score below 50 would be considered “Below the population average””

Comment: In certain sentences words such as ‘the’ and ‘a’ are missing (Refer to Rule 171: The word "the" should be added between the words "to" and WHO"s etc). 

Answer: Thank you for your comments. We have rephrased the sentence accordingly.

Line 174: “BMI was classified into 4 categories according to the WHO Asian – Pacific cutoff point: underweight (<18.5 kg/m2), normal weight (18.5–22.9 kg/m2), overweight (23–24.9 kg/m2), and obese (≥25 kg/m2) [32]”

Comment: COVID-19 is sometimes indicated as Covid-19 (Refer to Rules 101, 137, 156, 174 etc). 

Answer: Thank you for your comments. We have adjusted the sentences according to the recommendation to ensure consistency.

Comment: Other examples where attention needs to be paid to the editing of the manuscript involves the following: Rule 3: it should be students and not just "student" as it currently stands; Rule 136: the "who" needs to be added between the words "student" and "major".

It is strongly suggested that the manuscript should be language edited if it is to be reviewed again. 

Answer: Thank you for your comments. We have rewritten sections of the manuscript and hope that this complies with the reviewer’ remarks

---

## [Decision Letter · Decision Letter 1]

2 Oct 2022

PONE-D-22-14424R1Pattern and perceived changes in quality of life of Vietnamese medical and nursing students during the COVID-19 pandemicPLOS ONE

Dear Dr. Pham,

Thank you for submitting your manuscript to PLOS ONE. After careful consideration, we feel that it has merit but does not fully meet PLOS ONE’s publication criteria as it currently stands. Therefore, we invite you to submit a revised version of the manuscript that addresses the points raised during the review process.

The reviewers have engaged with your study and recommended some minor comments. Please address them as soon as you can.

We look forward to receiving your revised manuscript.

Kind regards,

Christmal Dela Christmals, PhD, MSc, BSc, RN

Academic Editor

PLOS ONE

Journal Requirements:

Reviewers' comments:

Reviewer's Responses to Questions

**Comments to the Author**

1. If the authors have adequately addressed your comments raised in a previous round of review and you feel that this manuscript is now acceptable for publication, you may indicate that here to bypass the “Comments to the Author” section, enter your conflict of interest statement in the “Confidential to Editor” section, and submit your "Accept" recommendation.

Reviewer #1: All comments have been addressed

Reviewer #3: (No Response)

2. Is the manuscript technically sound, and do the data support the conclusions?

Reviewer #1: Yes

Reviewer #3: Yes

3. Has the statistical analysis been performed appropriately and rigorously? 

Reviewer #1: Yes

Reviewer #3: Yes

4. Have the authors made all data underlying the findings in their manuscript fully available?

Reviewer #1: Yes

Reviewer #3: No

5. Is the manuscript presented in an intelligible fashion and written in standard English?

Reviewer #1: Yes

Reviewer #3: Yes

6. Review Comments to the Author

Reviewer #1: REVIEW COMMENTS

Pattern and perceived changes in quality of life of Vietnamese medical and nursing students during the COVID-19 pandemic

SUMMARY OF RESEARCH AND IMPRESSION

This study looked at the patterns and perceptions of changes in quality of care of medical and nursing students during Covid -19 pandemic. This study is really timely to identify issues that students face and how institutions can support them during training. This can also serve as a guide for replication of this study in other contexts. I recommend that the manuscript be accepted and published.

ABSTRACT

Abstract is well explained, and major findings highlighted

INTRODUCTION

Introduction has more information on medical students than nursing students although the study covers both specialties. I suggest you consider more informing pertaining to nurses.

METHODS

L140 – Check the sentence on this line use “was” instead of “is”

L152 – delete being from the sentence

Data collection instruments well explained

L211 – not having no change – correct the two negative words

L222 – can you provide a brief explanation the low response rate among the general Medicine students?

Statistical analysis well explained

RESULTS

Results well explained with required statistical methods

Tables are well labelled

DISCUSSION

Discussion covered the main findings of the study

Reviewer #3: Thank you for attending to the previous comments and recommendations that were made. There are however still a few places where COVID-19 is still referred to as Covid-19 or even covid-19. Refer to Lines 134; 140; 152 and 159. Line 233 starts with the word "Thorough" and it should be "Through". In Line 320 is should be Table 3 and not table 3. The same goes for Line 337 where it should be Table 4 and not table 4. The reason for the last two comments is because the Table elsewhere is written with a capital letter.

7. PLOS authors have the option to publish the peer review history of their article (what does this mean?). If published, this will include your full peer review and any attached files.

Reviewer #1: No

Reviewer #3: No

---

## [Author Response · Author response to Decision Letter 1]

10 Oct 2022

Dear Editors and Reviewers, 

Sincerest thanks for the previous editor’s and reviewers’ comments on our manuscript. 

We hope that a revised version of the manuscript will still be considered by PLOS One. We have modified the paper in response to the extensive and insightful reviewers’ comments. 

Furthermore, we have rewritten sections of the manuscript and hope that this complies with the reviewers’ remarks. We will respond to the comments point by point. 

All the line number mentioned in this document were based on the marked-up copy (the revised manuscript with track changes). 

Review Comments to the Author

Reviewer #1: REVIEW COMMENTS

Pattern and perceived changes in quality of life of Vietnamese medical and nursing students during the COVID-19 pandemic

SUMMARY OF RESEARCH AND IMPRESSION

This study looked at the patterns and perceptions of changes in quality of care of medical and nursing students during Covid -19 pandemic. This study is really timely to identify issues that students face and how institutions can support them during training. This can also serve as a guide for replication of this study in other contexts. I recommend that the manuscript be accepted and published.

Thank you for your comments!

ABSTRACT

Abstract is well explained, and major findings highlighted

Thank you for your comments!

INTRODUCTION

Introduction has more information on medical students than nursing students although the study covers both specialties. I suggest you consider more informing pertaining to nurses.

Thank you for your comments! We added several references according to your suggestion

“Healthcare workers were shown to have a significantly higher risk of COVID-19 infection, compared to the general public, and faced with extraordinary amounts of pressures leading to physical and mental exhaustion [9–14]” => We added 2 studies that look specifically at nurses in the front-line of COVID-19

“Medical and nursing students, before the pandemic, were already known to face multiple physical and mental problems including burnout, anxiety, depression, and other mental health issues, with stress, lack of academic motivation, and financial hardship being important risk factors [15–21]” => We added 2 studies that look specifically at nursing students

“Medical education also changed rapidly in response to the situation, with the replacement of in-person classes with online equivalents and disruption of clinical rotations [24–29]” => We added 2 studies that look specifically at nursing students and how their education was affected by COVID-19

METHODS

L140 – Check the sentence on this line use “was” instead of “is”

Thank you for your comments! We changed it to “is”

L152 – delete being from the sentence

Thank you for your comments! We deleted the word “being”

Data collection instruments well explained

Thank you for your comments!

L211 – not having no change – correct the two negative words

Thank you for your comments! We changed it to “having no change or improvement in their quality of life”

L222 – can you provide a brief explanation the low response rate among the general Medicine students?

Thank you for your comments! We added the explanation for this in line 462 in the limitation section

“The difference in response rate between academic majors could be due to clinical rotation’s scheduling, the willingness and interest of the class representatives, or miscommunications between the Office for Student Services and the class representatives. Due to the anonymous feature of the survey, we were not able to pinpoint the exact reasons for the difference in response rate and will look into this issue in future studies”

Statistical analysis well explained

Thank you for your comments!

RESULTS

Results well explained with required statistical methods

Tables are well labelled

Thank you for your comments!

DISCUSSION

Discussion covered the main findings of the study

Thank you for your comments!

Reviewer #3: Thank you for attending to the previous comments and recommendations that were made. There are however still a few places where COVID-19 is still referred to as Covid-19 or even covid-19. Refer to Lines 134; 140; 152 and 159. 

Thank you for your comments! We did a search for all inconsistencies in wording of COVID-19 and changed it as you suggested

Line 233 starts with the word "Thorough" and it should be "Through". 

Thank you for your comments! We changed it to "Through" as you pointed out

In Line 320 is should be Table 3 and not table 3. The same goes for Line 337 where it should be Table 4 and not table 4. The reason for the last two comments is because the Table elsewhere is written with a capital letter.

Thank you for your comments! We changed it to Table 3 and Table 4 as you pointed out

---

## [Decision Letter · Decision Letter 2]

7 Dec 2022

Pattern and perceived changes in quality of life of Vietnamese medical and nursing students during the COVID-19 pandemic

PONE-D-22-14424R2

Dear Dr. Pham,

We’re pleased to inform you that your manuscript has been judged scientifically suitable for publication and will be formally accepted for publication once it meets all outstanding technical requirements.

Kind regards,

Elsayed Abdelkreem, MD, PhD

Academic Editor

PLOS ONE

Additional Editor Comments (optional):

Reviewers' comments:

Reviewer's Responses to Questions

**Comments to the Author**

1. If the authors have adequately addressed your comments raised in a previous round of review and you feel that this manuscript is now acceptable for publication, you may indicate that here to bypass the “Comments to the Author” section, enter your conflict of interest statement in the “Confidential to Editor” section, and submit your "Accept" recommendation.

Reviewer #3: (No Response)

2. Is the manuscript technically sound, and do the data support the conclusions?

Reviewer #3: Yes

3. Has the statistical analysis been performed appropriately and rigorously? 

Reviewer #3: Yes

4. Have the authors made all data underlying the findings in their manuscript fully available?

Reviewer #3: Yes

5. Is the manuscript presented in an intelligible fashion and written in standard English?

Reviewer #3: Yes

6. Review Comments to the Author

Reviewer #3: ALL THE CORRECTIONS ARE SUFFUCIENTLY DONE AND I AM OF THE OPINION THAT IF THE EDITORS ARE OF THE OPINION THAT THE ARTICLE CAN BE PUBLISHED, THAT IT SHOULD BE PUBLIED

7. PLOS authors have the option to publish the peer review history of their article (what does this mean?). If published, this will include your full peer review and any attached files.

Reviewer #3: No

---

## [Editor Report · Acceptance letter]

14 Dec 2022

PONE-D-22-14424R2 

Pattern and perceived changes in quality of life of Vietnamese medical and nursing students during the COVID-19 pandemic 

Dear Dr. Pham:

I'm pleased to inform you that your manuscript has been deemed suitable for publication in PLOS ONE. Congratulations! Your manuscript is now with our production department. 

Kind regards, 

on behalf of

Dr. Elsayed Abdelkreem 

Academic Editor

PLOS ONE